**www.cambridge.org/ext**

Alien species; biodiversity; ecosystem services; ethics and policy; Novel Ecosystems

**Corresponding author:**
Christopher Lean;
Email: christopher.hunter.lean@gmail.com

# Open border ecosystems: Against globalised laissez-faire conservation

Christopher H Lean [ORCID]

Department of Philosophy, Macquarie University, Sydney, NSW, Australia and ARC Centre of Excellence in Synthetic Biology, Macquarie University, Sydney, NSW, Australia

## Abstract

Ecosystems are increasingly being represented as marketplaces that produce goods for humanity, and because of this, economic metaphors for increasing efficiency have been introduced into conservation. A powerful model for economic growth is the globalised free market, and some are implicitly deploying it to suggest changes in conservation practice. Ecological globalisation is the position that we should not control the free movement of species and rewilding occurs most efficiently through non-intervention. When species can move and interact with new ecological systems, they create novel ecosystems. These novel arrangements create experimental markets in nature's economy, providing opportunities for the efficient production of goods for humans, also known as ecosystem services. When invasive species supersede local populations, it indicates previous biotic systems were inefficient, which is why they were replaced, and therefore, it is wrong to protect indigenous "losers" from extinction. Those who defend indigenous species are accused of being xenophobic against recent biotic migrants. This position is flawed both empirically and morally as there is a disconnect between these economic and political arguments when applied to human economies and nature's economy.

## Impact statement

The adoption of ecosystem services as a goal in conservation has opened avenues for considering ecosystems as engines for economic production that can then be optimised through globalised free-market policy. Through open ecological borders and the global dispersal of species, new optimal arrangements will allow for new efficiencies, and arguments against such a policy are expressions of prejudice. These views have become increasingly promoted in the public sphere, finding support in the "new conservation" movement, among invasive species sceptics, animal rights proponents, the environmental humanities and economic free-market promoters. This theory transfer from economics is not justified as strong disanalogies exist between ecosystems and human markets. The application of globalised free-market policy to ecology excuses increased extinction, extirpation and a biotically homogenised world and so should be rejected.

## Introduction

The control of alien and invasive species has become increasingly contentious. The minority view that alien and invasive species should not be controlled has become increasingly prominent (Ricciardi and Ryan, 2018). Historically, the primary argument against the control of these species was on animal welfare grounds (Wallach et al., 2018). These arguments remain prominent but have been bolstered by novel arguments transplanted from socio-economic theory. I will describe and critique a novel position that represents the free movement of species into areas with no historical precedent as advantageous for humanity. This position is justified through analogy with globalised free markets in human economies. Invasive species cause extinctions, so this socio-economic justification for the free movement of alien species is often coupled with a case for species extinctions being acceptable as species preservation is inefficient.

These socio-economic arguments are largely normative rather than scientific, so they have not been forcefully argued in scientific papers, with some exceptions (Thomas, 2020). Rather they more commonly have appeared in popular science books and articles aimed at the public. Significant instances include the New York Times bestsellers, Fred Pearce's (2015) *The New Wild*, Christopher Thomas's (2017) *Inheritors of the Earth* and Emma Marris' (2011) *Rambunctious Garden.*[1] These arguments warrant significant enquiry as these views have dispersed into the public. I argue there is an underlying metaphor, based on global free-market socio-economic models, that suggests that we can optimise the efficiency of ecosystems through the free

---

[1] My discussion of Emma Marris's work is localised to Rambunctious Garden. Her later work has developed into positions that I believe are much more justified.

movement of species. Then, I will present some reasons we should not allow for the free movement of species and protect indigenous endemic species within their habitats.

## Ecology and economics

Ecology and economics have a long history of intellectual exchange, with models of optimal foraging, reproductive strategies, parental investment, altruism and many others being transferred across these sciences (Rapport and Turner, 1977).[2] It is unsurprising these disciplines have fluidly exchanged models. The phenomena of enquiry in both ecology and economics are structurally similar, the dynamics of populations and their response to resource allocations. Generally, this exchange has had some domain limitations. These exchanges were predominantly descriptive models rather than normative theories of how to optimise an ecosystem or market. Ecology's primary normative discipline, conservation biology, has historically been opposed to interventions that optimise nature's economy (unlike some sections of restoration and agroecology). Conservation has been driven by a normative code that looks to preserve natural systems or restore their dynamics to historical states (Lean, 2024). There have, however, been significant recent conceptual developments in the normative outlook of conservation biology: rewilding, ecosystem services and novel ecosystems. These developments have allowed for the introduction of normative socio-economic frameworks into conservation.

### Rewilding

Rewilding describes a range of conservation practices (Soulé and Noss, 1998; Carver et al., 2021). Two developments are significant for the normative intersection of economics and conservation. Rewilding can aim to create autonomous ecological systems, not beholden to the biotic history of that area. Alternatively, rewilding is a policy of passively allowing the encroachment of plants and wildlife onto abandoned land, allowing systems to self-organise (Regos et al., 2016). Both conceptualisations of rewilding feature the goal of the system becoming self-sufficient, without historical precedent dictating ecological compositions. The primary difference is whether rewilding is a result of human design or ecosystem self-organisation.[3]

### Ecosystem services

Ecosystem services are "the conditions and processes through which natural ecosystems, and the species that make them up, sustain and fulfil human life" (Daily, 1997). This is a directly anthropocentric representation of nature's value with the direct goal of translating nature's value into a framework amenable to economic valuation (Gómez-Baggethun et al., 2010). While the ecosystem services framework did not necessitate the economic valuation of nature, it fostered it. The Millennium Ecosystem Assessment identifies four types of service: provisioning (e.g. wood, food), regulating (e.g. water quality,

climate), cultural (e.g. recreation, aesthetic) and supporting (e.g. carbon cycle, soil formation) (Millennium Ecosystem Assessment, 2005).

### Novel ecosystems

Novel ecosystems are ecological arrangements without historical precedence (Hobbs et al., 2006, 2009). These are usually described as a passive rewilding process of species moving into areas disturbed through human actions. These novel ecological arrangements are often defended by arguing they provide ecosystem services (Evers et al., 2018).

The conceptual innovations in rewilding and ecosystem services provided the basis for novel ecosystems and their valuation in conservation theory. Novel ecosystems represent the value of non-historical ecological systems (Santana, 2022). The traditional view was that conservation is about conserving, maintaining, preserving, restoring and looking backwards for the reference states of what we should preserve. In novel ecosystems, we have a means for a forward-looking assessment. While ecosystems have often been analogised as economies or marketplaces, the direct economic representation of ecological systems via ecosystem services allowed some to consider ecosystems as economies that directly feed into human marketplaces. With the economic representation of ecosystems, derived from ecosystem services, there is the possibility of considering how ecosystems can be optimised for value. These preconditions allowed for the introduction of new economic normative frameworks to ecosystems. Within the socio-economic theory, one of the most powerful models (but not the only model) that has driven markets across the globe is the opening of markets globally and the liberalisation of trade through free-market principles. Unsurprisingly, this metaphor has implicitly been introduced into conservation, as I will now claim in more detail.

### Ecological globalisation

The socio-economic rhetoric of globalisation has been introduced into debates over conservation goals. In my description, these are a set of ideas that are pro-capitalist free market, with a strong emphasis on the globalisation of the marketplace and the deregulation of local markets, with a strong set of liberal social principles that include free movement across nation-state boundaries.[4] Rhetoric and metaphor have been redeployed from globalisation to ecology to establish that the free movement of capital and people across borders fosters innovative markets that yield more goods. Equally, there is the claim that expressions against such globalisation can be dismissed as veiled or open expressions of prejudice. I do not pass any judgement on these principles' application in the human domain. Rather, I am concerned with the application of these principles to ecological systems, and I emphasise the disanalogy between the biotic world and the human social world; claimed equivalencies I contend are a misguided anthropomorphising of the non-human world.

The application of this globalised economic perspective to ecology takes the following form. In contrast with traditional conservation, which looks to preserve species within their historic ecological communities and remove invasive species, ecological globalisation argues we should have an ecological globalised free market to produce services for humanity. They reject the protectionism of maintaining

---

[2]For example, Goodwin's class struggle model is an application of the Lotka–Volterra predator – prey model to explain wage growth (Goodwin, 1967).

[3]Self-organisation, in this literature and as I refer to it throughout the paper, is not a teleological process where there is an end state the ecosystem is designed to reach. Such views were common in early ecology under the lasting influence of 'the balance of nature'. Rather self-organisation is just the material conditions and interactions that structure the ecosystem over time.

[4]This could be described as neoliberalism, but the meaning and connotation of this term are so highly contested I have avoided using it.

existing ecological systems. Instead, they argue for more open biotic borders and reject the control of introduced species. Open borders are conjectured to allow for new biotic arrangements, which foster innovation. Innovation is identified with the novel ecological arrangements that populations create or in the adaptive responses of species to the novel ecological arrangements and abiotic environment. These innovations result in more productive communities, often with higher local species richness. Given that increased productivity and innovation provide humanity with more services, these ecosystems are more valuable than what previously existed. This is often coupled with a claim if endemic species are eliminated; they are "relics" that could not keep up with the modern ecological systems and should be allowed to die out. Arguments for the status quo are described as xenophobia against foreign species.

Without a commitment to preserving historical biotic arrangements, we are left with the question: "What is the goal of conservation?" The answer in much of this literature is for human economic development, which was forcefully argued by the "new conservationists" (Kareiva and Marvier, 2012; Kareiva et al., 2012). The environment is considered a vehicle for human economic development, so changes to these systems that profit local people through providing services are advantageous.

I will proceed to go through these different claims and identify proponents. I do not claim all the sources I discuss would assent to all the positions described above, but what I am concerned with, and proposing is the rhetoric that is being implicitly and explicitly introduced into conservation discourse. Differences in the package of views I describe result from the differing motivations of proponents.

The primary motivation for the introduction of this framework for ecosystem design is found in the growing movement against invasive species control. There has increasingly been a desire to represent invasive species as suffering from social injustice or that the existence of invasive species as a concept and the associated language around invasive species is an extension of human social injustice (Abbate and Fischer, 2019; Warren, 2021). Intellectual movements like multispecies justice have aimed to reframe conservation as an extension of human political practices and aspirations rather than primarily about preserving unique or historical biotic forms, fostering such interpretations (Celermajer et al., 2022). Equally compassionate conservationists have argued for animal personhood (Wallach et al., 2020). In these movements, the turn to these socio-economic arguments emerges from a rights-based argument about the individual rights of animals to have free movement and free association; this analogises these species to citizens in an economy who should be freely allowed to exercise their will in the local economy. To further justify this primary point, a background argument emerges: if the species do this, there will be positive implications for humanity as such free association will allow for more goods; as such, the free movement of species is both good for the species themselves and us.

The second source of support is those who wish to show the general applicability of these socio-economic principles. By showing that free markets are preferable not just in humanity but also in ecology, they show that these principles are more universally applicable. Further, there are direct advantages in the human domain. Protecting ecosystems from alien species is a major barrier to free trade, as there are strict biosecurity protocols that limit or increase the cost of trade across economies. Arguing that these barriers are unjustified would, some would conjecture, increase the efficiency of free markets. If ecological systems are more efficient when they are not governed, it would further the case for limited investment in conservation, reducing the need for governments to invest in conservation and reduce the public tax burden of conservation. There are, therefore, direct and ideological reasons for free-market proponents to defend an ecological globalised free market. This motivation is indicated by sympathetic articles and positive reviews for books espousing this ideology published by *The Economist* (e.g. 2017, 2022) and libertarian science writer Ronald Bailey (e.g. Bailey, 2000, 2010).

In the next section, I will break down these arguments into the claim that alien species increase services through species richness and innovation, that relic species should be allowed to die out and that biotic protectionism is xenophobic.

## Laissez-faire ecological globalisation

### Alien species increase services

> *Conservationists often suggest that protecting each last individual native species is somehow essential to maintaining … 'ecological services' … But that argument is a romantic illusion. Those services are best done by the species on hand that do it best. In much of the world that increasingly means nature's pesky, pushy invaders.* Pearce, 2015, 177

The reframing of conservation around providing services for local people has created an immense opportunity for the reappraisal of alien species. The local introduction of species is treated as equivalent to local species loss as "Facilitating the arrival of species… is just as legitimate—no more, no less—[as] an intervention in a dynamic system as managing existing biodiversity or attempting to avoid extirpations." (Thomas, 2020, 6). Or alien invasive species are presented as uniquely equipped to provide services, and given we want services, we should highly value invasive species (Pearce, 2015).

It is recognised that alien species provide significant ecosystem services for local people. Many species were introduced to provide services like animals introduced to hunt (deer, rabbits), trees for land reclamation (*Acacia, Pinus*) or flowers for their beauty (Scotch Thistle, Purple Loosestrife). Others have post hoc been recognised as providing services despite otherwise having significant negative impacts, such as the ability of zebra mussels to filter pollution from waterways (Thompson, 2014; Burlakova et al., 2023). Further, people's preferences change for species over time; Sagoff (2005) notes that Midwest wreath makers prefer the introduced oriental bittersweet over native species for making door ornaments. Such claims are legitimate, and significant work has gone into quantifying the contribution of alien species to human well-being, even in cases where these contributions have significant ecological trade-offs (e.g. de Carvalho-Souza et al., 2024).

The issue is whether alien species provision services to the extent that they should be promoted, or that we should remove protections against their introduction. Such a claim requires that there is a robust argument for alien species being better at providing services than the continued protection of indigenous species. Two lines of argument have been used to propose globalised ecologies produce more services. The first posits that the introduction of species across the globe will increase ecosystem services by increasing local species richness (see Lean, 2021) and the second argues invasive species drive biotic innovation, which will produce economic goods for humanity.

### Alien species are more productive

The introduction of alien species has often been claimed to increase biodiversity through increasing species richness (Marris, 2011;

Pearce, 2015; Thomas, 2017; Lundgren et al., 2024). This has been argued to lead to a more desirable and productive ecosystem; as Mark Sagoff (2005, 225) states, "If in any scientific (e.g. random) sample of ecosystems introduced organisms generally, overwhelmingly, and typically increase species richness, and if species richness supports desirable ecosystem properties, then one could argue these organisms benefit those systems." Given that biodiversity is often justified by its provision of ecosystem services, and therefore, the introduction of alien species will yield more goods for humanity.

This inference is largely justified by the Biodiversity-Ecosystem Services (BES) literature that sought to establish a firm relationship between biodiversity and economic output. To do this, initially, most early BES studies examined the relationship between local species richness and biomass production, under the assumption that local species richness represents biodiversity, and biomass production is a proxy for the economic utility of an ecosystem (Lean, 2021). In many field studies, more species were associated with more biomass and more biomass to more services, but the causal evidence for these relationships and their ability to be extrapolated is contested (Newman et al., 2017; Frank, 2022). Thus, this association between species count and biomass has been used to argue that if we continue introducing species, we will produce more service for humanity.

This general argument is bolstered, particularly in Pearce (2015), by the claim that the qualities that make alien species invasive are the same qualities that we desire for productivity. Invasion is associated with the displacement of native species through rapid population growth and competition. Increased growth rate, generation time, and dispersal will provide rapid biomass production and cycling across many ecological systems, which can be reinterpreted as productivity.

Equating ecosystem utility with the "production" of material quantities provides a natural bridge for the representation of ecological systems as markets producing goods. More species in an area, regardless of origin, leads to increased material outputs given that species richness increases ecosystem productivity.[5] These arguments mirror the methods of many countries to grow their economic production through migration. Introducing more people can increase the state's GDP even if the GDP per capita reduces. The argumentative structure within those arguing for a globalised ecology mirrors many economic arguments for globalisation in the human domain.

### Alien species for innovation

Traditionally, the ecosystem disruptions alien species caused were interpreted as a negative impact. New arguments have developed to reinterpret these impacts as a positive force driving biotic innovation. The innovations these disruptions are conjected to produce can be considered on two levels: (i) the forging of new relationships between species creates novel ecosystems, which can process resources in unexplored ways, and (ii) new interspecies relationships drive evolutionary innovation.

Unmanaged novel ecosystems "represent the wild lands of the future (i.e. the self-organised response of nature to anthropogenic impacts)." (Kueffer and Kaiser-Bunbury, 2014, 133). While many within the novel ecosystem literature, like Kueffer and Kaiser-Bunbury (2014), take this as a reason to assess the ecosystem for their trade-offs with traditional conservation targets; ecological globalists instead represent novel ecosystems as superior. Self-organisation in response to human disturbance is presented as

significantly more efficient than humanity's continuous management of ecosystems (Marris, 2011; Pearce, 2015).[6] While traditionally, environmental change is considered risky, with rapid change associated with environmental destruction, "new conservationists" emphasise that ecological systems are "resilient" and, therefore, can accommodate changes to create more human profit (Karieva et al., 2012). Novel ecosystems display the resilience of ecosystems in the face of change, so are better equipped to accommodate more change, ultimately providing more opportunities for human development (Marris, 2011; Pearce, 2015).

Novel ecosystems are natural experiments in the organisation of populations. These experiments then reach self-stable arrangements, from which we can identify profitable goods. We should allow for experiments in ecological arrangements as:

- The species introduced are often introduced because they are already adapted to human-affected environments, so intrinsically they are resilient to human changes.
- The overall arrangements are a natural response to human-induced environmental change, making them resilient to future changes.
- Many of these species were originally introduced around the globe as they had some economic use, so they may project further economic gains as they are introduced into new regions and possibly become more efficient and profitable through combining with new species.
- Unfilled functional roles resulting from human-induced damages can be filled through species introductions. This fills the gaps created by local extinctions, allowing for ecosystem repair.

Novel ecosystems are directly analogised with human cosmopolitanism, where immigration and new combinations of cultures can foster economic and cultural innovation (Raffles, 2011; The Economist, 2017; Pineda-Pinto et al., 2023). Or as Keulartz and Van der Weele (2009) state in their plea for a reframing of invasion biology: "the mixing and blending of cultural identities…lead to new forms of diversity" (101) and suggest an alternative framing of conservation where "the inevitability of the mixing up of ecologies in a globalised world… does not necessarily lead to a loss of diversity: mixing up individual species will lead to new patterns and new local systems" (114). There is a repeated analogy between the human domain – where immigration can aid the economy through the reorganisation of people, capital, cross-cultural borrowing and the creation of new goods and markets – and ecosystems. The thought is that ecosystems will similarly gain from novelty introduced from across the globe.

Innovation is also described as emerging through evolutionary change. Alien species create new selection pressures, competing and consuming local species. This skews the inheritance of both local and introduced species to create novel adaptations to the environmental conditions they create. The introduction of the Cane toad to Australia has caused the death of many of Australia's small predators throughout the regions of their spread but this has driven change. Snake species have evolved smaller heads and larger bodies, meaning that they are less likely to swallow a cane toad, and if they do, they will have more mass to survive the poison (Phillips and Shine, 2004). Equally, cane toads have evolved to exploit the opportunity that the Australian continent affords them, evolving

---

[5]Strong evidence exists that invasive species cause massive ecosystem service loss, so the empirical evidence for these claims is wanting (Walsh et al., 2016).

[6]Given this argument, a natural conclusion would be that the conscious creation of novel ecosystems would also be considered less efficient as it is not the result of natural ecological self-organisation but human management. But Pearce, for instance, praises designed novel ecosystems, particularly Ascension Island (Pearce, 2015).

longer legs to hop farther and find new environments (Hudson et al., 2016). This rapid evolution occurred since the cane toads' introduction in the 1930s. This is just an instance of the wide range of adaptive radiations that occur with species introduction. Introduced species can display "invasive adaptations" such as increased growth rate, dispersal ability (like the cane toad) and shorter generation times (Whitney and Gabler, 2008).

Likewise, the rapid evolution of local species in response to introduction is contextualised in terms of a changing globe. If indigenous species adapt to invasion, they may also become robust to other human-induced changes, such as land clearing, climate change and pollution. Alien species are then seen as training indigenous species for the globalised world through their disruptions. Or as Pearce states (2015, 211), "By seeking only to conserve and protect the endangered and the weak, it becomes a brake on evolution and a douser of adaptation. If we want to assist nature to regenerate, we need to promote change, rather than hold it back."

Those sympathetic to ecological globalisation argue that the rapid evolution afforded by such interspecies confrontations is desirable and creates new biodiversity better adapted to the environments humanity has created. Thomas argues that we need to take a long-term perspective on conservation and consider how our actions will affect biodiversity in millions of years rather than the near future (Marris, 2011; Thomas et al., 2022). Hyperabundant species that spread across the globe will be the basis for further adaptive radiations in the long term (Thomas, 2017). Abundant populations have more chances for the creation of mutations, and the spread of the species means they will be exposed to many new environments. This could provide a basis for many different locally adaptive subspecies of globally distributed species, which could then evolve into the new species of future ecosystems.

### Withdraw investment in "relic" and "loser" species

The preference for alien and invasive species that are conjectured to drive innovation and provide services is coupled with a dismissal of endangered indigenous species. These species are described as "relics" or "losers" and are argued to not warrant significant investment for their conservation. Given these populations are in decline, they are viewed as incapable of adapting to the changing world we are creating (Thompson, 2011; Kareiva and Marvier, 2012; Pearce, 2015; Thomas, 2017, 2020). Instead of investing in their preservation, we should accept their eventual extinction as the long-term future lies with populations that can adapt to our degraded natural world. These populations are invasive species whose high abundance and adaptations to human-degraded environments make them highly capable of surviving. By denigrating preserving extant biodiversity as "nostalgia" for "the world as it once was," they present a view of progress where extinction is a morally neutral phenomenon (Kareiva and Marvier, 2012).

Such positions are justified when considered under the rubric of a species' contribution to ecosystem productivity. The free-market Reason (2000) magazine and the new conservationists Kareiva et al. (2012) make near-identical statements that even the loss of the previously highly abundant American chestnut had little effect on ecosystems.[7] The case for losing endangered species is even stronger

---

[7]"In many circumstances, the demise of formerly abundant species can be inconsequential to ecosystem function. The American chestnut, once a dominant tree in eastern North America, has been extinguished by a foreign disease, yet the forest ecosystem is surprisingly unaffected." Kareiva et al., 2012 and "The loss of American chestnuts was economically damaging, but the ecological costs are much less clear." (Bailey, 2000).

for these authors as the small population sizes of endangered species limit their ecosystem effects and they are often functionally extinct in their habitat (Thompson, 2011). They, therefore, cannot currently contribute to ecosystem services, in addition to lacking the population size and adaptive features that would allow them to evolutionarily respond to changing environments. Equally, in line with the economic argument for reducing conservation investment, it could be more economically viable to allow local extinctions to happen and then introduce new species to maintain ecosystem function; "it could be less costly and more practical to introduce new elements that thrive under the new conditions than attempt to save the last few individuals of species that will inevitably die out from that location" (Thomas, 2020, 6).

The futility of conservation in the modern world is often emphasised, in contrast with the arguably more feasible project of preserving ecosystem services. The political scientist Stephen Meyer (2006) states this plainly in "The End of the Wild." He argues that although we have lost the race to save biodiversity, we must save the ecosystems that provide us with services. Throughout the book, he describes species as relics or ghosts of a world that is now completely lost. This sort of fatalism about the fate of species provides the backdrop for those arguing for a turn to preserving economically significant ecological systems rather than preventing extinctions. This type of fatalistic language has, horrifically, historically been applied to indigenous people. Indigenous Australians were described as a "dying race" through the late 19th and early 20th century, justifying the withholding of support and the removal of their children (McGregor, 2015).

The adoption of the free-market language around economic winners and losers in this literature is stark. In economics, there is a common argument that instead of investing or supporting small companies, or even industries, affected by globalisation, they should be left to go bankrupt to create a more efficient market, regardless of the effects on the local people in an area. Of course, like in the conservation case, allowing "relics" to disappear is a decision made by those in power rather than an inevitability. A social Darwinian interpretation of economics is being applied to ecosystems, where the hungry, dynamic, disruptive alien species upend the stale relics, creating more efficient ecosystems. Like lassie-fare economics, the goal is overall functionality and market efficiency. If a business with an essential service goes bankrupt due to a temporarily volatile market, a new business will emerge to serve that role. Equally, in ecology, if a necessary ecosystem function is lost, an alien species can be reintroduced to fill that role. This assumed fungibility between businesses, or equally that the people negatively affected are equivalent to those people who later gain, is similarly applied to species. An endemic species loss is treated as equivalent to an introduced species gain.

### Conservation is xenophobic

One can object to allowing species to go extinct by claiming that they prefer these endemic endangered species. This preference may be born from these species providing a sense of identity and uniqueness to their local landscape (Hettinger, 2021). Such preferences are commonplace among the public (e.g. Tait et al., 2017). Critics of indigenous species conservation respond that these preferences are an expression of xenophobia. The debate increasingly mirrors socio-economic arguments around human migration, where opposing migration is labelled as xenophobia. In the human context, workers and unions often oppose high migration because it increases competition for jobs or housing. In response, their

opponents may accuse them of xenophobia. Similarly, in the biological context, defending indigenous species is sometimes portrayed as xenophobic against the free movement of other biological populations.

This concern often appears in animal-rights-derived defences of invasive species (e.g. Winograd, 2013). Daniel Ramp, director of the Centre for Compassionate Conservation in Sydney, argues Australia's feral cat program is based on unexamined stigmas towards invasive species and "xenophobia (Aguirre, 2019)."[8] Often these arguments are coupled with direct analogies with human migration and cosmopolitanism (Raffles, 2011). Sonia Shah (2020) draws a book-length analogy between human immigration and the movement of species. Some go as far as to draw comparisons between ecologists and right-wing ideologies to discredit current ecological science. For instance, Peretti (1998) attempted to link conservation science with Nazi and apartheid governments, thereby questioning its validity.

This argument emerged from justified concerns within ecology about the problematic language used to describe invasive species. Terms like "invasive," "alien," "foreign," and "interloper," often coupled with militaristic language, were seen as inappropriate and rooted in a troubled history (Larson, 2005). Subramaniam (2014, 105–106) identifies the particularly capricious way invasive species are represented in the media with headlines like "It's a Cancer" and "The invasion of the woodland soil snatchers." Ecological metaphors of displacement have supported segregation in urban planning (Cresswell, 1997). Ecologists recognised that these metaphors were misleading, and leading figures called for correcting the language used (Brown and Sax, 2004).

Brown and Sax (2004) state: "It is **not** to argue that exotic species are good so that their spread should be fostered. This is **not** to suggest that modern humans should let nature take its course and elect not to intervene in the dynamics of dispersal and extinction. It is to plead for more scientific objectivity and less emotional xenophobia." (my emphasis).

This plea for objectivity asks for a revision of language rather than an endorsement of invasive species. This is different from the current movement that frames conservationists who prioritise native species as xenophobic. If we take the above statement by Brown and Sax and remove the "nots" from their quote, we gain the modern position advocating for the spread of exotic species and suggesting that humans should let nature take its course. This interpretation argues that the language scientists use reflects irrational prejudice, undermining the validity of the current scientific research. Further, given that anti-xenophobic politics often leads to more open borders between nation-states, open borders are the correct policy for ecological systems (this appears to be the implicit argument in Shah, 2020). These shifts from correcting rhetoric to applied policy in ecosystems are distinctly new developments in the debate over conservation norms.

## Reasons for scepticism towards ecological globalisation

### Problems with services

There has been a strong movement towards restructuring conservation, turning it away from preserving nature for its own sake and towards using conservation for human benefit (Kareiva et al., 2012; Kareiva and Marvier, 2012). When benefit to humanity is narrowly

construed as leading to economic development, it will inevitably boil down to profit and actively changing ecosystems to increase profit (Kareiva et al., 2012). Ecological globalisation is one interpretation of how to increase the profitability of ecosystems by directly representing them as marketplaces that would benefit from free-market principles such as open borders, radical experimentation and a lack of safety nets, which, in turn, will produce services for humanity.

Ecosystem services could be justified as being nature directed and not just human centred through the belief that ecosystem services increase when biodiversity is high. So even if we solely desire nature for its economic utility, we are required to preserve biodiversity. The problem is that this connection is not as strongly evidenced as many would like (Newman et al., 2017; Frank, 2022). If biodiversity is only valuable for profitable services, ultimately, biodiversity is unnecessary. The studies of BES do not establish that extant biodiversity is better equipped than designed ecosystems or monocultures to provide goods and services (Newman et al., 2017). To start with the evident, agriculture involves monocultures and is highly productive and lucrative. We need additional value bases for preserving biodiversity other than its immediate productivity, or designed systems will be preferable, and conservation will cease to be an endeavour.

There is, of course, a wealth of additional reasons to preserve biodiversity. These exist even within the ecosystem services framework when it is not narrowly construed. Ecosystem services include cultural and recreational services, which can be inclusive of the aesthetic value of the natural world. While these services are often overlooked in empirical studies (Boerema et al., 2017), they offer alternative ways to conceive of ecosystem value rather than that of economies that must be optimised for productivity or profit. Rapid change in ecosystems and extinction can diminish cultural and recreational value as we break the connection between an ecosystem and its local people (Hettinger, 2001, 2021). To consider only nature as valuable in terms of its immediate economic utility misses the incredibly diverse values applied to it.

The overemphasis in some ecosystem services research on economic valuation has led to conceptual change in environmental policy. The Intergovernmental Science-Policy Platform on Biodiversity and Ecosystem Services (IPBES) has pressed that ecosystem services must be developed to include Nature's Contribution to People (NCP). These are "all the contributions, both positive and negative, of living nature (diversity of organisms, ecosystems, and their associated ecological and evolutionary processes) to people's quality of life" (Díaz et al., 2018, 270). This extremely encompassing concept was envisioned primarily to focus attention on non-economic valuations, especially on the way local people value the environment in diverse and culturally unique ways. As a result, there are efforts to cement non-economic interpretations of ecosystem services or extensions to the ecosystem services framework. The question is whether and how such conceptual innovation can or will be implemented on the ground.

Within environmental ethics, many consider the environment valuable, regardless of its instrumental use (McShane, 2007). This can be strongly defined as an objective value that exists, even when a valuer is not present or weaker as the fact that people value something without considering its utility (Morrow, 2024). Weak intrinsic value is like existence value, where agents value something existing in the world regardless of their interaction with it (Attfield, 1998). Existence value was developed in ecological economics, a field of economics much less sympathetic to the free-market theories found in other economic subfields that have been adopted by invasive species supporters.

---

[8]According to this perspective, efforts to protect indigenous species from invasive feral cats are based on xenophobia against cats in a country with an estimated 5 million pet cats.

Similarly, heritage and aesthetic values are not derived from the utilisation of ecosystems other than observing them, both first hand and through their depiction in media. Aesthetic value can be derived from the immediate pleasure of experiencing nature or through the appreciation and enjoyment that coming to learn about these entities provides (Welchman, 2020). Heritage value is derived from an entity's history, producing cultural and intellectual significance (Thompson, 2000). It is best understood through analogy; just as the temples of Angkor Wat have heritage value through their ability to connect current people to human history and inspire awe, natural systems provide a window into natural history. The development of ecosystems and their radical alteration undermines these values as rapid change alters their historical features or loses unique features to the more ubiquitous variation found globally.

There are other ways to conceive of non-use values or values that do not demand integration into economic productivity (e.g. the land ethic [Millstein, 2024] or deep ecology [Diehm, 2020]). Such values provide an alternative to conceiving ecosystems as engines for development, as development ignores what many people value about having a world that is not just human beings and their immediate interests. Placing human development first is extremely detrimental to conservation. If addressing human inequalities is a precondition for biodiversity protection, protecting the natural world will remain indefinitely at the end of our to-do list (Kopnina et al., 2018). Standardly, it is accepted that conservation can trade off with economic development (e.g. Hirsch et al., 2011). This is expected when there are competing values involved and people value nature in varied ways. Considering ecosystems as markets for service production will ultimately justify little but the replacement of biotic systems with more productive designed human systems.

### Homogenisation is not diversity

Ecological globalisation does appear to increase local species richness (alpha diversity). Numerous datasets indicate that while species richness may not be in decline, or even increasing in areas, beta diversity seems to be declining as a limited selection of generalist species replaces endemics (e.g. Nowakowski et al., 2018; Lewthwaite and Mooers, 2022). Beta diversity is a measure of the extent to which the addition of some unit of biodiversity (e.g. species, ecosystem type, genes, functions) adds something new to a regional pool. This addition of new characters is what is often valued in diversity, rather than a simple count of objects, and is closely linked to the protection of biotic features from extinction. A rare or endemic feature or species will naturally add something new to a regional count. The gains in alpha diversity found in local systems due to common species being added results in less beta diversity (Kortz and Magurran, 2019). This is the long-recognised global homogenisation of ecosystems (McKinney and Lockwood, 1999; Sax et al., 2002; Olden et al., 2005).

There are several reasons we may want a high emphasis on uniqueness in our diversity measures. Unique species in themselves contain features that provide opportunities for the future. These opportunities are described as option value. Retaining unique biotic options is a prudent bet-hedging strategy to account for future uncertainty, particularly given their loss is irreversible (Arrow and Fisher, 1974; Faith, 1992; Lean, 2017). These uses may be material, such as for medicine or biotechnology, or could be newfound cultural or aesthetic ways to appreciate organisms. As such diversity not only connects to current instrumental use but to maintaining a set of possibilities for future engagement. Homogenisation adds to the count of individuals present but does not create new options. This is not just true due to ubiquitous species being added across the globe while rare species disappear, it is also a result of what types of species are being lost or introduced.

Extinction and introductions are not random. Over 50% of species are in significant decline; in contrast, only 1–2% are invasive, and another 5–29% of species are increasing their ranges (McKinney and Lockwood, 1999). There is a global trend for the introduction of generalist species, which can survive well under a wide set of conditions and exploit a wide range of resources, while losing specialist species, that excel in exploiting difficult-to-utilise resources. This trend is argued to lead to a global functional homogenisation (Clavel et al., 2011). The loss of specialists is significant for biodiversity as they have highly coordinated adaptations, producing novel morphologies and biochemical profiles. The loss of novelty in species features is a loss for the potential future engagement with these species. Functional homogenisation not only reduces the variation of the features of species but also the range of biotic interactions that occur in the world (Valiente-Banuet et al., 2015). Globalisation of ecology leads to the extinction of not only species and features but also processes, reducing the complexity and variety of biotic arrangements that could exist.

One common reply to current diversity losses is to argue that the long-term diversification of ubiquitous species will compensate for the short-term losses (Marris, 2011; Thomas, 2020). The push to consider conservation as only assessed by its impact on the distant future is deeply immoral. Evolution is a slow process; the recovery from a mass extinction event takes millions of years. An ecosystem's functional recovery from a mass extinction event is estimated to take two million years and species recovery takes 10 million years (Alvarez et al., 2019). These time scales make the recovery morally irrelevant. We are making conservation decisions for humans alive now and in the near future or our societies as they continue into the more distant future. In 10 million years, the current human societies and even the human species will not be around to benefit from such a policy. Two million years of unstable ecosystems will create a persistent negative impact on whatever sentient life will exist if we allow for a mass extinction event. The extended time for recovery disanalogies' biological diversity from cultural diversity. Cultural change is rapid, often taking less than a generation (Boyd and Richerson, 2005). The cosmopolitan human cultures analogised in this literature diversify rapidly, unlike biotic diversity. Further, recovery is not morally salient if you believe the natural world can be harmed. If you cause grievous bodily harm to someone and they recover in 5 years through surgery and significant physical therapy, you have still harmed them.

Ultimately, the wider the base of species that we allow to exist now, the more diverse the species that will exist in the distant future. Through aggressively pruning the tree of life, there will be fewer branches from which to sprout new lifeforms. There is a wealth of studies that show the depth and breadth of loss when endemic unique species go extinct (e.g. Davis et al., 2018). Invasive species are a major cause of the decline and extinction of endemic species; just invasive predators have "been implicated in 87 bird, 45 mammal, and 10 reptile species extinctions—58% of these groups' contemporary extinctions worldwide" (Doherty et al., 2016), and there is a strong indication that invasive species are the second largest cause of recent extinction (Bellard et al., 2016). Future diversification is not an adequate compensation, or as Pauchard et al. (2018, 2) state: "Even if one is willing to offset the current losses of biodiversity with the promise of new biodiversity as non-native species evolve and diverge, millions of years of biological adaptation and evolutionary history would be lost."

Homogenisation and the loss of unique lineages are antithetical to any act to preserve biodiversity and preserve the range of values that diverse biotic entities hold.

### Anthropomorphising ecosystems as markets

There is a long history of biology being used to justify the worst of human actions. Concepts of "alpha males" and "bad genes" have been used to justify grave injustice in human society. The solution to such narratives has been to show that both these interpretations of biology are wrong and, *more importantly*, that such reasoning is an extension of the naturalistic fallacy. We cannot read from nature to the way humans should be. Whether we are taking biological exemplars that correspond to or clash with our political beliefs, we cannot infer from the way biology is the way we should be.

Normative belief should not be dictated by or dictate the structure of non-human biology.[9] Just because we are a Rawlsian, communitarian, communist or free-market capitalist, it does not mean ecosystems should be. The social, legal, historical and material differences between ecosystems and human socio-economic entities, particularly those of nation-states, mean significant disanalogies exist. The capacities of the natural world are determined by their causal and structural affordances. While ecosystems self-organise like markets, human markets contain human agents who can envision possible future market structures and possibilities, a potential not found in ecosystems. Human cognition, communication, and culture allow for rapid responses to disruption; evolution is much slower (See Table 1). Even if globalised migration is a positive force in the socio-economic standing of our society, it does not mean that we are justified in exporting this policy to nature. What is morally right needs to be possible; we need to understand the actual capacities of biological systems before radically changing them, and it needs to be justified on ethical principles that account for the different normative and material considerations of conservation.

In the debate over whether invasive species is a xenophobic concept, we can accept that there is a legitimate discussion of whether the term itself and references to invasive species could be rephrased to reduce capricious misuse. But this does not discredit the position that we should prevent introduced species from displacing endemic species, given the normative importance of extinction (Wienhues et al., 2023). We should take seriously that there is a difference between the values involved in determining whether species extinction is acceptable and the values for whether international migration between nation-states is acceptable. Supporting the removal of xenophobic language does not necessitate a position where one applies "anti-xenophobic" politics to conservation policy.

As such, I am not arguing that the social movements that have inspired these new positions are wrong. One can assent that xenophobia is bad and/or global free markets are good. However, treating ecosystems as an extension of such human socio-economic systems is a gross overreach. Many argue that there is no justification for the separation of humanity and nature in ecology (e.g. Inkpen, 2017) but to treat nature as just an extension of humanity and its interests pulls to another extreme. Especially,

---

[9]As a moral naturalist, I hold biology is significant for ethics; it provides constraints for our psychological and social capacities. Ethics is, however, derived from social coordination, influenced by evolved normative reasoning and historically entrenched norms, and ongoing rational and social deliberation. The deeply social aspect of morality means non-human organisms, who do not have such complex social and communicational capacities, cannot determine our ethics.

**Table 1.** Analogy between globalised markets and globalised ecosystems

| Category | Globalised economies | Globalised ecosystems | Issues with analogy |
|---|---|---|---|
| Goals and means | Growth in economic production is the goal of a market, and efficiency is the means. | Human development through the supply of ecosystem services is the goal. Efficient cycling of resources and biomass production the means. | Other values include social, cultural, intrinsic, aesthetic, land ethics, existence value, deep ecology, wilderness, heritage value and biodiversity. |
| Migration | Free Migration derived from human rights. | Free migration derived from animal rights. | Human rights are not the same as animal rights e.g. a right to political expression is absurd in animals. |
| Regulation | Deregulated local markets self-organise to create efficiency. | Reduce management and allow for ecosystems to self-organise to create efficiency. | Management preserves endemic species, stops expatriations and results in functional/less volatile ecosystems. |
| Innovation | New combinations of people and ideas through cosmopolitanism produce new goods and efficiency. | New combinations of species produce more services. | Cultural evolution, communication and rational (or considered) deliberation allow for coordination in the human case. |
| Policy | Outdated businesses should be allowed to fail. | Relic species should be allowed to fail. | Replacing businesses takes years, and replacing lost species takes millions of years. |
| Response to opposition | Opposition to migration is xenophobic. | Opposition to introduced species is xenophobic. | Correcting the language and application of these terms does not necessitate stopping the protection of endemic or rare species. |

when this belief is a vehicle for justifying the extinction of unique biotic forms, withdrawing investment in conservation, and a means of rejecting precaution in conservation.

### Conclusion

Finding the value in the biotic arrangement created since human globalisation is important. Novel ecosystems contribute to conservation

goals by supplying ecosystem services or habitats for rare species. There is, however, a difference between accepting the value in the changed world and advocating for changing the world's biota. There are, unfortunately, more voices raising the metaphor of an open-border free-market ecology. Others have recognised this style of argument emerging in the conservation literature. John Halley (2019, 1451), in his review of *Inheritors of the Earth*, states: "I had a feeling of deja vu with this argument. It is the same one used by the promoters of globalisation in the early 90s. In those days, it was suggested that globalisation represented some inexorable flow of history that could not be resisted." This free-market globalisation metaphor is used to justify withdrawing funding for controlling invasive species and to represent endemic species as "losers" whose protection is inefficient. Efficiency is presented as morally right, as we need productive ecosystems for human development. Throughout this discourse, overly economic interpretations of ecosystem services have been a pernicious force in conservation thinking by implicitly rejecting the asymmetry of preserving versus creating biotic features and connecting ecosystems directly to economic productivity (Lean, 2024). This paper has aimed to clarify this style of argument in the hope of making conservationists aware and hopefully wary of it. Ecosystems are not economic markets for us to optimise; open biotic borders will lead to extinctions; the extinction of a species now is not counterweighted by species gained in millions of years, and nature does not solely exist for our economic needs.

**Open peer review.** To view the open peer review materials for this article, please visit http://doi.org/10.1017/ext.2025.10002.

**Data availability statement.** There is no data for this comment.

**Acknowledgements.** I would like to thank the Philosophy of Science reading group at Macquarie University for feedback on a draft of this paper and to all the audiences that have seen versions of this paper in presentation form.

**Author contribution.** I am the sole author.

**Financial support.** Christopher Lean, during the research of this paper, received funding from the Australian Government through the Australian Research Council Centres of Excellence funding scheme (project CE200100029). The views presented in this paper are the author's alone and do not necessarily reflect those of his funders.

**Competing interests.** I am unaware of any competing interests that would bear on the content of this paper.

**Ethical approval.** There were no experiments or surveys conducted for this paper. Therefore, formal ethics approval was not required. Ethical academic standards were maintained throughout the production of the paper.

**Informed consent.** Informed consent does not apply to this publication.

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
