## [Reviewer Report]

I really like this paper, even though I disagree with much of it.

The paper takes seriously the theoretical parallels between economic globalization and ecological globalization, and argues that our optimal attitudes from one domain don’t carry over to the other because of key differences, summarized nicely in Table 1.

While I’d debate some of the details, those details are ongoing controversies, and not reasons to hold back on publishing the paper.

I have a few suggestions, however.

First is in the setup. Contra Dr. Lean, I don’t think it’s true that economics is mostly normative while ecology is mostly descriptive, with the notable exception of conservation bio. Both ecology and economics are diverse disciplines, and economics has all sorts of subfields that aim to be descriptive (e.g. behavioral economics and econometrics), and ecology has many applied branches that are intentionally normative (e.g. restoration ecology and agroecology).

Second, this matters, because the paper often glides over whether the relevant difference-making features are empirical or political, and how compelling the argument is depends on this. For example, on p.16, Lean points to dimensions of value in nature that markets don’t track. He seems to infer from this that Ecosystem Services approaches also won’t represent these values. But they (and similar approaches like Nature’s Benefits to People) could, with the right measurement tools—including ones found in applied branches of economics like environmental economics.

Put briefly, I find compelling that there are empirical differences between ecosystems and markets which make theory transfer between domains risky. Less compelling is the claim that there are value differences between ecology and economics, given the diversity of those fields. The values in conservation bio aren’t the same as in restoration eco or invasion biology. The values in health economics aren’t the same as in game theory.

I’d like to see Lean sort this out a bit more.

---

## [Reviewer Report]

Overall, I found this to a very interesting analysis and critique of work by the “new conservationists.” The analysis describes an analogy between their work on topic such as invasive species and novel ecosystems and that of global free markets. It ends with a critique of the analogy and views of these new conservationists. The critique asserts that ecosystems are not markets and treating them as such reduces their values to ones that can be priced in terms of human preferences and invasive species homogenize ecosystems. Though I found the arguments powerful, I will mention few places were I think the analysis and argument can be strengthened.

First, the author writes that “while ecology is ostensibly a descriptive science; economics is a normative science...” This is true but I would add that many ecologists have thought that there is a ‘balance of nature’. So, in some sense, they have treated communities and ecosystems as normative. It would be helpful to distinguish more carefully these two sources of normatively.

Second, the author does note in places that ecosystem services need not be anthropocentric. It is really important to note that they need not be economic either. An inference from ecosystem function/service to pricing them is not the same thing. We shouldn’t treat them as if they are.

Third, novel ecosystems are ones in which new species arrive, colonize, and restructure those very systems. However, the author often speaks as if these processes “self-organize”. My sense of the discussion of novel ecosystems is that it is as much about humans “designing” them as their own assembling. It might be helpful to mention this point.

Finally, the author writes, "While ecosystems self-organise like markets, human markets

contain human agents who can envision possible future market structures and possibilities, a potential not found in ecosystems." Unfortunately, this sounds like a human/nature dualism which is implausible post-Darwin. Humans are after all parts of ecosystems too. One might argue that humans are primates with culture. Most importantly, we are able to take moral responsibility for our actions in ways that no other animal can. It is this perfectly natural property that differentiates us from the rest of nature.

So, I think this paper should be published with minor revisions.

---

## [Reviewer Report]

I’m entirely satisfied with the revisions. The paper is now more clear, and avoids the overgeneralization that hampered the previous draft.

---

## [Reviewer Report]

Having read and commented on this essay before, I am satisfied that all my questions have been adequately addressed. I think it is ready for publication.

---

## [Editor Report]

Thank you for the revision. Both reviewers are now satisfied with the revisions, and I am happy to recommend acceptance of R1. Please also accept my apologies for the delay here. The original Handling Editor was unexpectedly unable to response, which caused an unfortunate block in the final decision. This has now been resolved.